# Antibody responses to SARS-CoV-2 in patients with differing severities of coronavirus disease 2019

Ekasit Kowitdamrong[1,2], Thanyawee Puthanakit[3,4], Watsamon Jantarabenjakul[3,4,5], Eakachai Prompetchara[6,7], Pintip Suchartlikitwong[1,4], Opass Putcharoen[5,8], Nattiya Hirankarn [1,9] *

1 Department of Microbiology, Faculty of Medicine, Chulalongkorn University, Bangkok, Thailand, 2 Applied Medical Virology Research Unit, Chulalongkorn University, Bangkok, Thailand, 3 Department of Paediatrics, Faculty of Medicine, Chulalongkorn University, Bangkok, Thailand, 4 Center of Excellence in Pediatric Infectious Diseases and Vaccines, Chulalongkorn University, Bangkok, Thailand, 5 Thai Red Cross Emerging Infectious Diseases Clinical Centre, King Chulalongkorn Memorial Hospital, Bangkok, Thailand, 6 Center of Excellence in Vaccine Research and Development (ChulaVRC), Chulalongkorn University, Bangkok, Thailand, 7 Department of Laboratory Medicine, Faculty of Medicine, Chulalongkorn University, Bangkok, Thailand, 8 Department of Medicine, Faculty of Medicine, Chulalongkorn University, Bangkok, Thailand, 9 Center of Excellence in Immunology and Immune-mediated Diseases, Chulalongkorn University, Bangkok, Thailand

* Nattiya.H@chula.ac.th

**Data Availability Statement:** All relevant data are within the manuscript and its Supporting Information files.

## Abstract

### Background

A greater understanding of the antibody response to SARS-CoV-2 in an infected population is important for the development of a vaccination.

### Aim

To investigate SARS-CoV-2 IgA and IgG antibodies in Thai patients with differing severities of COVID-19.

### Methods

Plasma from the following patient groups was examined: 118 adult patients with confirmed SARS-CoV-2 infections, 49 patients under investigation (without confirmed infections), 20 patients with other respiratory infections, and 102 healthy control patients. Anti-SARS-CoV-2 enzyme-linked immunosorbent assay (ELISA) from EUROIMMUN was performed to assess for IgA and IgG antibodies. The optical density (OD) ratio cutoff for a positive result was 1.1 for IgA and 0.8 for IgG. Additionally, the association of the antibody response with both the severity of disease and the date after onset of symptoms was analyzed.

### Results

A total of 289 participants were enrolled and 384 samples analyzed from March 10 to May 31, 2020. Patients were categorized, based on their clinical manifestations, as mild (n = 59), moderate (n = 27), or severe (n = 32). The overall sensitivity of IgA and IgG from the

**Funding:** This work was supported by funding to support Biobank from Ratchadapisek Sompoch Fund, Faculty of Medicine, Chulalongkorn University.

**Competing interests:** The authors have declared that no competing interests exist.

samples collected after day 7 of the symptoms was 87.9% (95% CI: 79.8–93.6) and 84.8% (95% CI: 76.2–91.3), respectively. Compared to the mild group, the severe group had significantly higher levels of spike 1 (S1) antigen-specific IgA and IgG. All patients in the moderate and severe groups had S1-specific IgG, while 20% of the patients in the mild group did not have any IgG detected after two weeks after the onset of symptoms. Interestingly, in the severe group, the SARS-CoV-2 IgG level was significantly higher in males than females ($p$ = 0.003).

## Conclusion

The serological test for SARS-CoV-2 has a high sensitivity more than two weeks after the onset of illness. Additionally, the serological response differs among patients based on sex as well as the severity of infection.

## Introduction

In late December 2019, an outbreak of initially undiagnosed pneumonia was reported in Wuhan, Hubei Province, China [1]. The causative pathogen was later identified as a novel beta coronavirus closely related to the severe acute respiratory syndrome (SARS) coronavirus (CoV) family and was recently termed SARS-CoV-2 [2]. As of July 30, 2020, more than 17 million people were infected with SARS-CoV-2, and there were up to 670,000 SARS-CoV-2-associated deaths [3]. The first case in Thailand was reported on January 12, 2020 and was a traveler from Wuhan [4]. On July 30, 2020, there were 3,304 confirmed SARS-CoV-2 cases in Thailand, with an epicenter in the Bangkok metropolitan area. Real-time reverse transcription polymerase chain reaction (RT-PCR) diagnostic assays are a goal standard for case ascertainment and diagnosis [5]. However, validated serological tests provide evidence to compliment virological diagnoses, particularly in or after the second week of infection [6]. A greater understanding of the antibody response in an infected population is beneficial for the development of a vaccine.

Enzyme-linked immunosorbent assay (ELISA) is commonly used to access viral-specific antibodies in a quantitative manner, and for decades has been widely accepted as a diagnostic test for antibodies. The sensitive, quantitative measurements of ELISA make it suitable to assess dynamic changes in viral-specific antibodies. In principle, antigen-specific IgM and IgA should be detected in approximately the second week of infection, followed by antigen-specific IgG after the second week of infection. There are several serology platforms currently available, which use various antigens. One large nucleocapsid-based ELISA study assessing 208 samples reported that IgM and IgA were detected 3–6 days after the onset of symptoms with a sensitivity of 85.4% and 92.7%, respectively, while IgG was detected later, 10–18 days after the onset of symptoms, with a sensitivity of 77.9% [7]. Interestingly, another study showed that the seroconversion if IgG against the SARS-CoV-2 nucleocapsid and a peptide from the spike region was detected as early as that of IgM and reached its peak within six days after seroconversion [8]. Compared to patients with severe cases, a weaker and more rapidly declining antibody response was observed in asymptomatic patients and in those with milder symptoms [9].

The EUROIMMUN anti-SARS-CoV-2 ELISA was one of the first CE-marked (European Conformity) diagnostic assays developed and available worldwide. It assesses the response of IgA and IgG to the spike 1 (S1) protein and has been reported to correlate well with the plaque reduction neutralization test (PRNT) [10, 11]. The EUROIMMUN IgG assay received

Emergency Use Authorization (EUA) from the United States (US) Food and Drug Administration (FDA). Thus far, most of the results have been reported from Europe and the US. The objective of this study was to investigate the response of IgA and IgG antibodies to SARS-CoV-2 in serial blood samples collected from a population of Thai patients with confirmed COVID-19, and the association of these responses with the severity of the illness.

## Materials and methods

The present study was conducted at the Thai Red Cross Emerging Infectious Diseases Clinical Center (TRC-EIDCC) and the Faculty of Medicine at Chulalongkorn University. The study present was reviewed and approved by the Institutional Review Board of the Faculty of Medicine (IRB number 242/63) and the National Blood Center, Thai Red Cross Society (COA No. NBC 5/2020).

### Patient population

Confirmed COVID-19 cases were defined as those that tested positive for SARS-CoV-2 RNA using real-time reverse transcription-polymerase chain reaction (RT-PCR) testing of combined nasopharyngeal and throat swab (NT) samples. RT-PCR testing was performed in the Department of Microbiology of the Faculty of Medicine at Chulalongkorn University. SARS-CoV-2 RNA was detected using the cobas® SARS-CoV-2 kit (Roche Diagnostics, Basel, Switzerland) on a fully automated cobas® 6800 system (Roche Diagnostics, Basel, Switzerland) according to the manufacturer's recommendations. Nucleic acid was automatically extracted from 400 μL of the NT specimens in viral transport medium (VTM) along with added internal control RNA (RNA IC). Subsequent real-time RT-PCR was performed automatically by the system, targeting ORF1a/b and E genes specific to SARS-CoV-2 and pan-Sarbecovirus, respectively.

Classification of the confirmed case was as follows, according to the COVID-19 management guideline of the Thai Ministry of Public Health: 1) mild–asymptomatic or upper respiratory tract infection (URI), 2) moderate–pneumonia without hypoxia, and 3) severe–pneumonia with hypoxia, of which the antiviral treatment was given. The date of the onset of symptoms, disease severity, hospitalization time, and personal demographic information were obtained from hospital medical records. The control group included three subgroups. The first subgroup included 20 plasma samples collected from healthy volunteers in the laboratory and 82 plasma samples leftover from healthy blood donors prior to February 2020. The second subgroup included 49 plasma samples collected from May 1 to May 31, 2020, from patients under investigation (PUI) for COVID-19 with RT-PCR results that were negative for SARS-CoV-2. The third control subgroup included 20 serum specimens collected from May 1 to May 31, 2020 from patients with other infections (Dengue, HBV, HCV, HIV, Mumps, Measles, Rubella, EBV, CMV, VZV, HSV, and Treponema). Plasma and serum were aliquoted and stored at -20˚C prior to serological testing.

### Laboratory methods

Plasma samples of 10 μL were diluted to 1:101 in sample buffer in order to perform SARS-CoV-2 S1-specific IgA and IgG assays using anti-SARS-CoV-2 ELISA IgG/IgA kits (Euroimmun, Lubeck, Germany) according to the manufacturer's instructions. Semi-quantitative results were evaluated by calculating the ratio of extinction at 450 nm of each sample over the calibrator. A cutoff ratio of 1.1 was used for SARS-CoV-2 IgA, as suggested by the package insert. The borderline cutoff ratio of 0.8 for SARS-CoV-2 IgG was assigned as positive.

## Statistical analysis

Demographic characteristics were described for each patient. Continuous variables were expressed as the median with an interquartile range (IQR). Differences in continuous and categorical variables between the two groups were assessed using the Wilcoxon rank-sum test and Chi-square test or the Fisher exact test, respectively. Sensitivity, specificity, positive predictive value (PPV), and negative predictive value (NPV) were also calculated.

## Results

### Demographics of the population

There were 118 confirmed SARS-CoV-2 infections from March 10 to May 31, 2020: 59 with mild (upper respiratory symptoms), 27 with moderate (pneumonia without hypoxia), and 32 with severe (pneumonia with hypoxia), with a median age of 38 years (IQR: 27–48). A total of 213 samples collected from 118 patients were tested for antibodies against SARS-CoV-2, with 36 patients having 1 sample, 69 patients having 2 samples, and 13 patients having 3 samples. A total of 99 samples were collected seven days after the onset of symptoms. There were 49 PUI who were negative for SARS-CoV-2, with a median age of 47 years (IQR: 28–65 years), 25 males and 24 females. The baseline clinical characteristics are summarized in Table 1. There were significant differences in age and sex between the groups, with the patients in the severe group being mostly male (66%) and 40–59 years old.

### Seroconversion of antibodies against SARS-CoV-2 in COVID 19 patients

Among the 118 confirmed SARS-CoV-2 patients, 99 had blood samples collected at least once more than 7 days after the onset of symptoms. The overall seroconversion of antibodies after the 7[th] day of symptoms is summarized in Table 2. The overall sensitivity of IgA was 87.9% (95% CI: 79.8–93.6) with a negative predictive value of 93.1% (95% CI: 88.3–96.4). The overall sensitivity of IgG was 84.8% (95% CI: 76.2–91.3) with a negative predictive value of 91.0% (95% CI: 87.9–96.1). The overall specificities of IgA and IgG were 94.7% and 97.1%, respectively.

The specificity, however, varied between the control subgroups. The raw data for all controls are shown in S1 Table. There were two false-positive IgA and one false-positive IgG results out of 102 healthy controls. We were able to obtain and re-analyze seven samples, with an OD ratio $\geq$ 0.8 after two months. The OD ratio was quite similar to the initial results, confirming the healthy control group's limited background. Of the 49 PUI, there were five positive IgA and two positive IgG results for COVID-19 with negative RT-PCR results for SARS-CoV-2. Two of these patient's tests were repeated after 2–4 weeks, and the OD ratios returned to

**Table 1. Clinical characteristics of patients.**

| | COVID (N = 118) | Mild (N = 59) | Mod (N = 27) | Severe (N = 32) | Non-COVID (N = 49) | P-value |
|---|---|---|---|---|---|---|
| Median Age (IQR) | 38 (27–48) | 29 (26–39) | 39 (27–47) | 49 (41–58) | 47 (28–65) | <0.001 |
| Age group, n (%) | | | | | | <0.001 |
| • < 20 | 6 (5) | 4 (7) | 1 (4) | 1 (3) | 3 (6) | |
| • 20–39 | 61 (52) | 41 (70) | 14 (52) | 6 (19) | 17 (35) | |
| • 40–59 | 43 (36) | 12 (20) | 11 (40) | 20 (62) | 14 (28) | |
| • > 60 | 8 (7) | 2 (3) | 1 (4) | 5 (16) | 15 (31) | |
| Male, n (%) | 47 (40) | 19 (32) | 7 (26) | 21 (66) | 25 (51) | <0.001 |

**Table 2. The overall sensitivity of samples collected after the 7[th] day of symptoms.**

| ELISA_IgA ≥ 1.1 | n/N | % | 95%CI | |
|---|---|---|---|---|
| Sensitivity | 87/99 | 87.9 | 79.8 | 93.6 |
| Specificity[a] | 162/171 | 94.7 | 90.2 | 97.6 |
| Positive predictive value | 87/96 | 90.6 | 82.9 | 95.6 |
| Negative predictive value | 162/174 | 93.1 | 88.3 | 96.4 |
| ROC area (Sens. + Spec.)/2 | - | 0.91 | 0.88 | 0.95 |
| ELISA_IgG ≥ 0.8 | | % | 95%CI | |
| Sensitivity | 84/99 | 84.8 | 76.2 | 91.3 |
| Specificity[b] | 166/171 | 97.1 | 93.3 | 99 |
| Positive predictive value | 84/89 | 94.4 | 87.4 | 98.2 |
| Negative predictive value | 166/181 | 91.7 | 87.9 | 96.1 |
| ROC area (Sens. + Spec.)/2 | - | 0.91 | 0.87 | 0.95 |

[a] IgA Specificity in Healthy control = 100/102 = 98.03%, Specificity in Patients under investigation for COVID-19 with SARS-CoV-2 RT-PCR negative = 44/49 = 89.8%, Specificity in cross-reactivity panel group = 18/20 = 90%.
[b] IgG Specificity in Healthy control = 101/102 = 99.01%, Specificity in Patients under investigation for COVID-19 with SARS-CoV-2 RT-PCR negative = 47/49 = 95.9%, Specificity in cross-reactivity panel group = 18/20 = 90%.

normal. Of 20 serum specimens collected from patients with other infections, there were two samples with both IgA and IgG cross-reactivity to CMV- and EBV-positive samples.

## Seroconversion of antibodies, stratified by day of illness and disease severity

The seroconversion of the antibodies, stratified by the day of illness, is shown in Table 3. The sensitivity for serological testing within seven days of the onset of symptoms was only 29.7–30.6% for IgA and 10.2–16.2% for IgG. The IgA positivity rate increased to 60% during the 2nd week and 100% during the 3rd-4th weeks, and then declined to 81.9% in the 2nd month. The IgG positivity rate increased to 90% during the 3rd-4th weeks of diseases.

**Table 3. The seroconversion of antibody stratifies by day of illness and severity (N = 213 tests).**

| | PCR positive | ELISA IgA ≥ 1.1 | | | |
|---|---|---|---|---|---|
| | | Total | Mild | Moderate | Severe |
| Day 0–3 | 37 | 11/37 (29.7%) | 6/23 (26%) | 3/9 (33.3%) | 2/5 (40%) |
| Day 4–7 | 49 | 15/49 (30.6%) | 3/25 (12%) | 7/10 (70%) | 5/14 (35.7%) |
| Day 8–14 | 45 | 27/45 (60%) | 8/20 (40%) | 9/13 (69.2%) | 10/12 (83.3%) |
| Day 15–28 | 21 | 21/21 (100%) | 4/4 (100%) | 5/5 (100%) | 12/12 (100%) |
| Day > 28 | 61 | 50/61 (81.9%) | 22/31 (71%) | 14/15 (93.3%) | 14/15 (93.3%) |
| | 213 | 124/213 (58.2%) | 43/103 (41.7%) | 38/52 (73.1%) | 43/58 (74.1%) |
| | PCR positive | ELISA IgG ≥ 0.8 | | | |
| | | Total | Mild | Moderate | Severe |
| Day 0–3 | 37 | 6 (16.2%) | 1/23 (4.3%) | 3/9 (33.3%) | 2/5 (40%) |
| Day 4–7 | 49 | 5 (10.2%) | 0/25 (0%) | 1/10 (10%) | 4/14 (28.6%) |
| Day 8–14 | 45 | 14 (31.1%) | 4/20 (20%) | 4/13 (30.8%) | 6/12 (50%) |
| Day 15–28 | 21 | 19 (90.5%) | 2/4 (50%) | 5/5 (100%) | 12/12 (100%) |
| Day > 28 | 61 | 55 (90.2%) | 26/31 (83.9%) | 14/15 (93.3%) | 15/15 (100%) |
| | 213 | 99 (46.5%) | 33/103 (32%) | 27/52 (51.9%) | 39/58 (67.2%) |

To investigate the association of antibody levels to the severity of the disease, the antibody levels at the first time point were expressed using the specified cutoff value, stratified by disease severity. The severe group had a significantly higher level of S1-specific IgA and IgG antibodies compared to the mild group (Fig 1). It should be noted that the two patients in the severe group who did had no detectable S1-specific IgA were tested only once, at 31 and 40 days after the onset of symptoms, and therefore it was likely that the IgA levels had already declined in these patients.

To see the dynamics of each group, we plotted the average antibody level from mild, moderate, and severe groups at five intervals (Fig 2).

There were 103, 52, and 58 samples from the mild, moderate, and severe groups, respectively (Table 3). A clear pattern emerged, showing that the severe and moderate groups had significantly higher IgA and IgG levels 15 days post-symptoms compared to the mild group. Of the group with mild symptoms, 20% (7/35) of the samples had no detectable IgG antibodies more than 2 weeks after the onset of symptoms. Only 1 out of 15 patients from the moderate group had no detectable IgG antibodies, while all 15 patients with severe symptoms had high IgG levels after the second week (Table 3).

Since age and sex were associated with the disease outcome, we analyzed the correlation between antibody level and age in the severe group, as shown in Fig 3A and 3B; however, no significant correlation was found.

We also compared the antibody levels between males and females within the severe group. Interestingly, the levels of both S1-specific IgA and IgG to were higher in males than in females, with IgG being statistically significant (Fig 4A and 4B). The median age of males (51, IQR: 43–59) was also higher than that of females (41, IQR: 24–46) in the severe group.

## Discussion

The results of the present study have demonstrated that during the first week of COVID-19 infection, the sensitivity of the antibody response to acute viral infection is low. The antibody

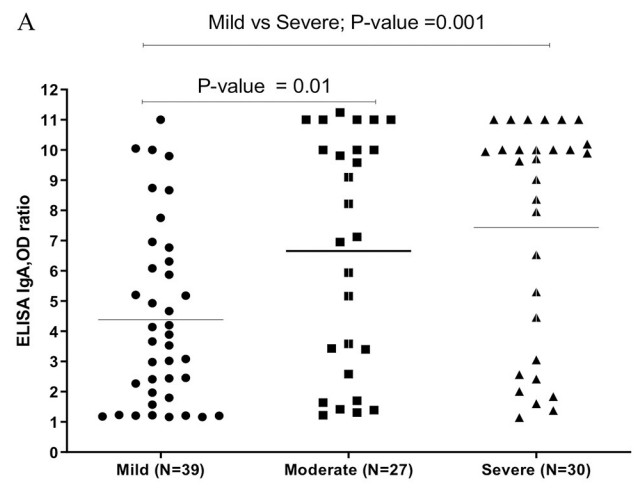

| ELISA IgA ≥1.1 | Mild | Moderate | Severe |
|---|---|---|---|
| 96/118 | 39/59 (66%) | 27/27 (100%) | 30/32 (94%) |

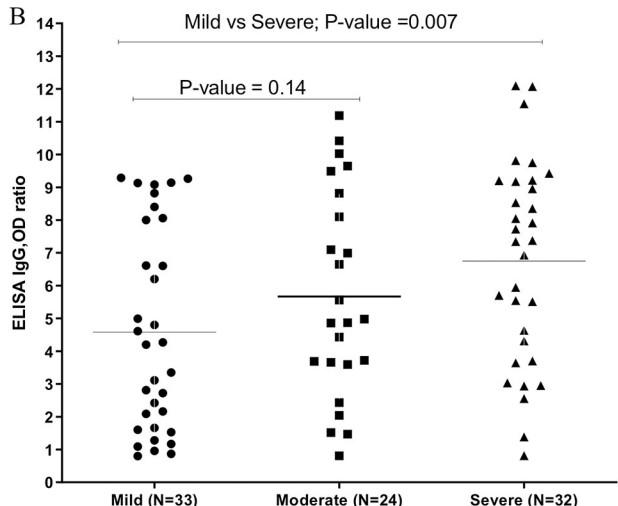

| ELISA IgG ≥0.8 | Mild | Moderate | Severe |
|---|---|---|---|
| 89/118 | 33/59 (56%) | 24/27 (89%) | 32/32 (100%) |

**Fig 1.** Antibody levels based on disease severity: A) ELISA IgA OD ratio, B) ELISA IgG OD ratio.

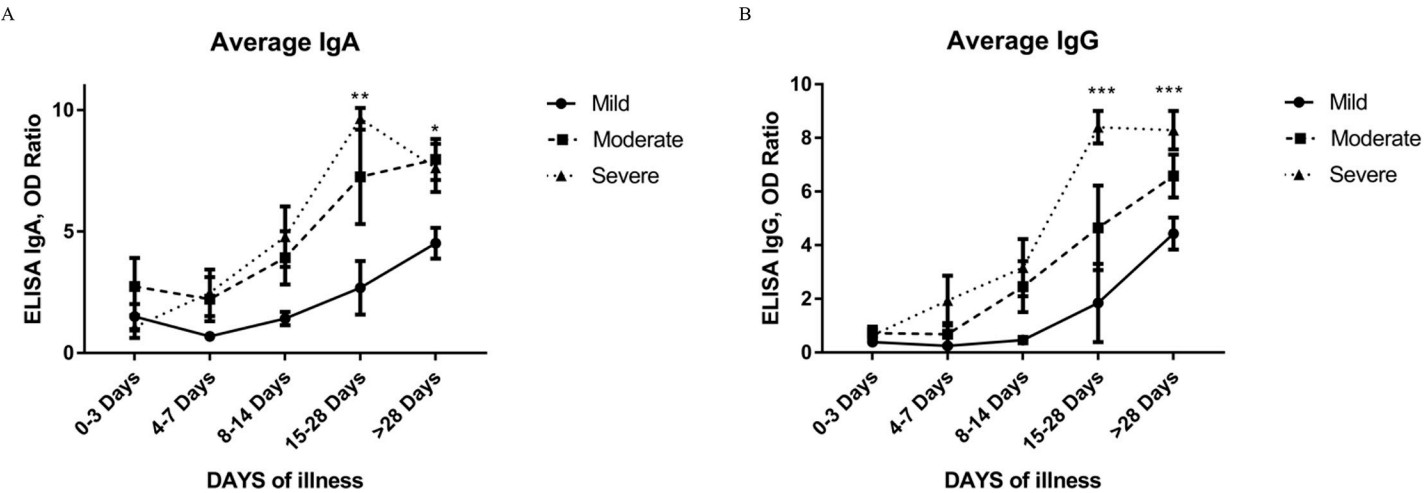

**Fig 2.** Average antibody levels among COVID-19 patients with different disease severity by date of illness: A) IgA OD ratio and B) IgG OD ratio.

response seen in the present study started with IgA, followed by IgG. Because it is difficult to compare results using different serological analyses, we have only used data from studies that tested with EUROIMMUN for comparison. The results from 15 studies using EUROIMMUN assays are summarized in S2 Table. Previous studies have mostly reported that the sensitivity of IgA within the first week was less than 60% [12, 13]. In the present study, 30% of COVID-19 patients developed positive IgA antibodies very early, within 3 days after the onset of symptoms. Therefore, the presence of positive IgA antibodies might help identify some COVID-19 patients in the early stage, however, negative results cannot be used to exclude infection. The seroconversion of IgA was 100% in 21 patients at 15–28 days after the onset of symptoms. In a study from France, a 100% sensitivity of IgA seroconversion was reported in 82 cases after the second week of symptoms [13], and in 91 patients after the third week [14]. Interestingly, in the present study we noticed a decline in IgA after one month, with the sensitivity decreasing to 80%.

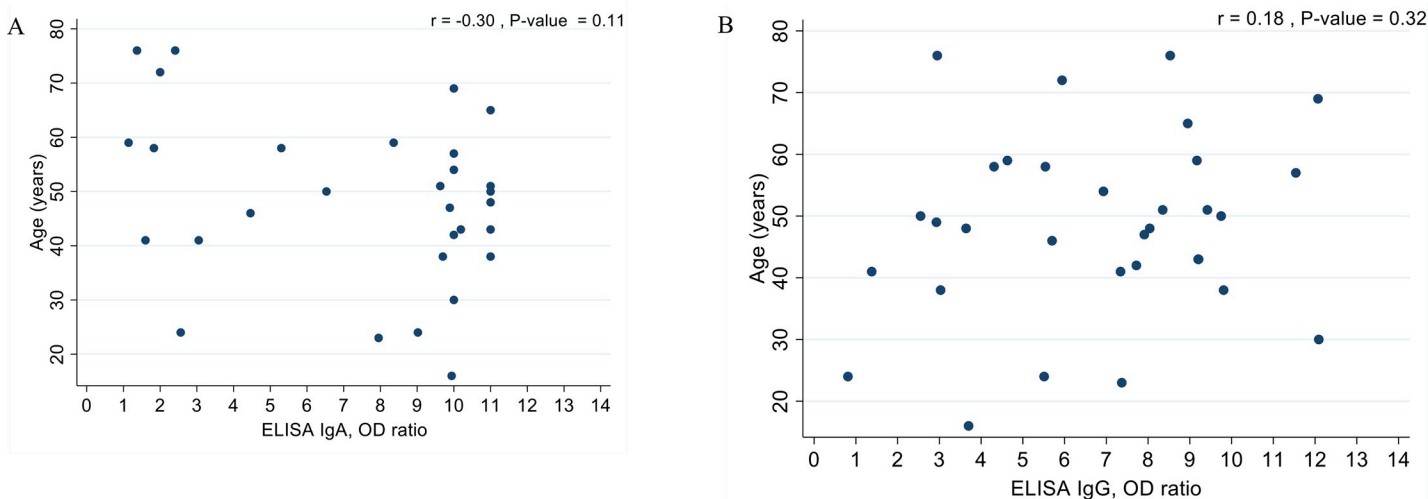

**Fig 3.** The correlation between antibody level and age in the severe group: A) Age VS ELISA IgA OD ratio, B) Age VS ELISA IgG OD ratio.

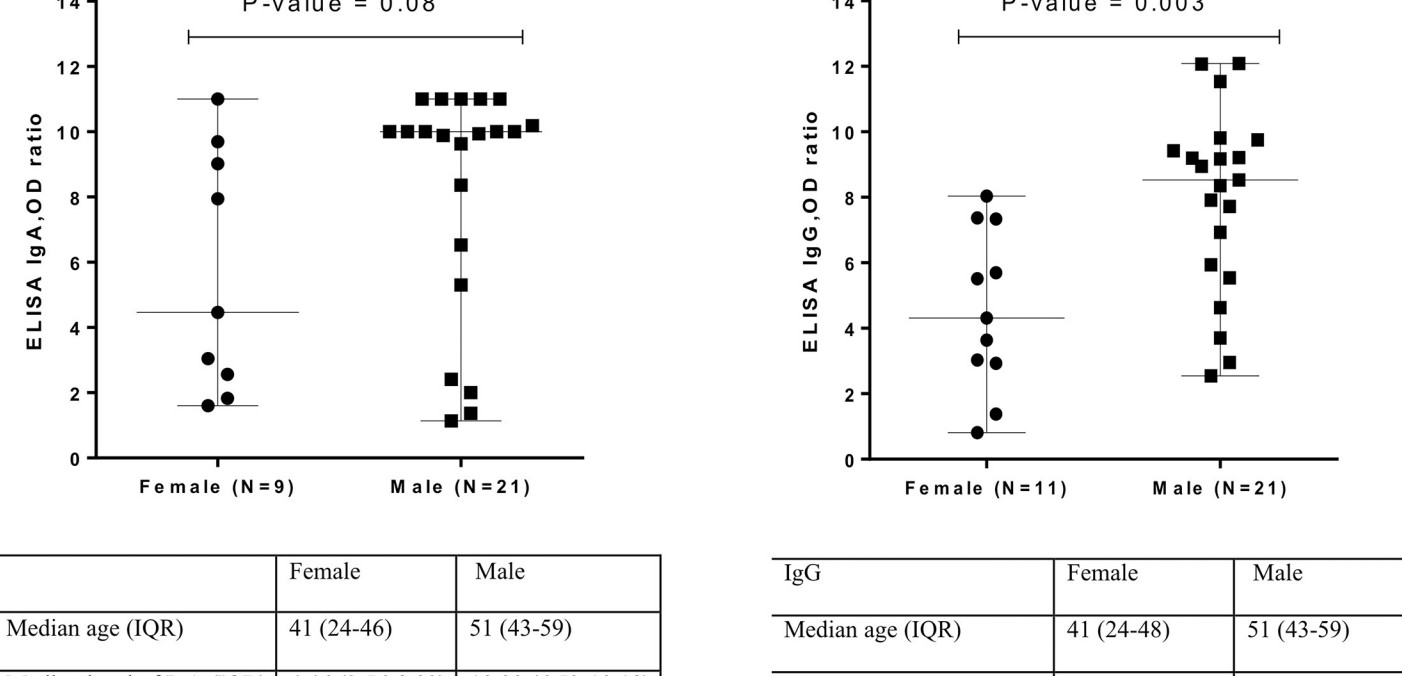

**Fig 4.** The relationship between antibody levels and sex in the severe group: A) Sex VS ELISA IgA OD ratio, B) Sex VS ELISA IgG OD ratio.

In regard to IgG, it should be noted that for the present study, we used the borderline cutoff to define a positive result to increase the sensitivity of IgG. This cutoff did not change the specificity of the test. IgG antibodies specific to the SARS-CoV-2 S1 antigen developed later than IgA. The sensitivity of IgG was 90% after the second week of symptoms, which is comparable to other studies [13, 15–17].

In the present study, 20% of the patients with mild symptoms did not develop any IgG antibodies specific to COVID-19, even after 2 weeks after the onset of symptoms. Other studies have found up to 20–30% of cases to be negative for IgG [18, 19]. When we analyzed the correlation of antibody levels with clinical severity, it was clear that patients with more severe clinical manifestations had higher antibody levels, for both IgA and IgG, than patients in the mild group. This observation has been consistently reported in other study populations as well [10, 19, 20]. The explanation behind these findings is not yet clear, however, one current hypothesis is that the elevated inflammatory response in the severe patients might produce a more robust immune response, including antibody production from B-lymphocytes. It also raises concerns about the role of antibody-mediated severity, although there is no evidence to support it.

Moreover, several studies have reported that there were higher rates of severity and mortality in male patients [21]. In the present study, more females were infected with COVID-19 than males (60% female vs. 40% male); however, there were significantly more males (66%) in the severe group. Interestingly, we found a significantly higher level of IgG in males than in females in the severe group, similar to recent results found by Klein et al. [19]. The median age of males was higher than that of females in the severe group. It is possible that higher levels of antibodies might be associated with greater illness severity in male patients. However, there is a speculation that biological sex might affect immunity through various mechanisms [21]. Although women seem to have greater antibody responses and are more susceptible to

autoimmune diseases than men, other factors, including innate immunity, regulatory T cells, expression of angiotensin-converting enzyme 2 (ACE2), or other mechanisms related to sex hormones might explain the greater severity and higher antibody levels observed in male patients. Further studies are needed to elucidate the impact of sex on disease severity, which might lead to a better understanding of this challenging disease.

Although the assessment of specificity was not the main objective of this study, our results confirmed those from previous studies, that the specificity of EUROIMMUN anti-SAR-CoV-2 IgA is lower than that of IgG. As summarized in S2 Table, the specificity of EUROIMMUN anti-SAR-CoV-2 IgA ranged from 68.3–94.6%, while anti-SAR-CoV-2 IgG ranged from 85–100%. A higher background was observed in the control group with respiratory symptoms. As previously stated, of the 49 PUI, there were 5 positive IgA and 2 positive IgG results that had corresponding negative RT-PCR results for SARS-CoV-2. Two of these patients were repeated after 2–4 weeks, and the OD ratio returned to normal. It should be noted that the positive results in these patients could be the result of either a false positive or a true positive case with negative RT-PCR results. However, there was no evidence to support COVID-19 infection in these patients. We also observed two samples with both IgA and IgG cross-reactivity with CMV- and EBV-positive samples. Since we did not have serological results for other coronaviruses in these two samples, we did not know definitively if they were directly cross-reactive with CMV and EBV, or the result of cross-reactivity with another coronavirus. However, based on results from other studies summarized in S2 Table, there were reports of false positives with various infections, including EBV and CMV seropositives [12, 17].

In summary, the present study extensively reported the serological responses of COVID-19 patients in Thailand up to 60 days after the onset of symptoms. Although most of the samples were tested at two time points, blood samples were collected from patients at different stages and at various intervals. Therefore, we did not determine a median time for positive results, which might have been subject to bias.

## Supporting information

**S1 Table. Raw data of control group.**
(DOCX)

**S2 Table. Summary serological results from 15 studies based on Euroimmun test.**
(DOCX)

## Acknowledgments

We would like to thank the health care team for at King Chulalongkorn Memorial hospital, Thai Red Cross, particularly Dr. Kampol Suwanpimolkul, Dr. Leilanee Paitoonpong, and Dr. Suvaporn Anulgulreungkitt. Special thanks for the advice from Dr. Parvapan Bhattarakosol and statistical analysis by Miss Jiratchaya Sophonphan.

## Author Contributions

**Conceptualization:** Ekasit Kowitdamrong, Thanyawee Puthanakit, Nattiya Hirankarn.

**Data curation:** Ekasit Kowitdamrong, Thanyawee Puthanakit, Pintip Suchartlikitwong, Opass Putcharoen, Nattiya Hirankarn.

**Formal analysis:** Ekasit Kowitdamrong, Thanyawee Puthanakit.

**Funding acquisition:** Nattiya Hirankarn.

**Investigation:** Ekasit Kowitdamrong, Opass Putcharoen.

**Methodology:** Ekasit Kowitdamrong, Eakachai Prompetchara.

**Project administration:** Nattiya Hirankarn.

**Resources:** Watsamon Jantarabenjakul, Eakachai Prompetchara, Pintip Suchartlikitwong, Opass Putcharoen.

**Supervision:** Nattiya Hirankarn.

**Validation:** Watsamon Jantarabenjakul, Pintip Suchartlikitwong, Opass Putcharoen.

**Writing – original draft:** Ekasit Kowitdamrong, Nattiya Hirankarn.

**Writing – review & editing:** Thanyawee Puthanakit, Watsamon Jantarabenjakul, Eakachai Prompetchara, Pintip Suchartlikitwong, Opass Putcharoen, Nattiya Hirankarn.

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
