## [Decision Letter · Decision Letter 0]

14 Sep 2020

PONE-D-20-23838

Antibody Responses to SARS-CoV-2 in Coronavirus Diseases 2019 Patients with Different Severity

PLOS ONE

Dear Dr. Hirankarn, 

Thank you for submitting your manuscript to PLOS ONE. After careful consideration, we feel that it has merit but does not fully meet PLOS ONE’s publication criteria as it currently stands. Therefore, we invite you to submit a revised version of the manuscript that addresses the points raised during the review process.

We look forward to receiving your revised manuscript.

Kind regards,

Walter R. Taylor

Academic Editor

PLOS ONE

Additional Editor Comments:

Dear Dr. Hirankarn,

we have received comments from one reviewer.

The decision is minor revision but could you try to find a native English speaker to improve the language, please?

yours sincerely,

Walter Taylor.

Journal Requirements:

Reviewers' comments:

Reviewer's Responses to Questions

**Comments to the Author**

1. Is the manuscript technically sound, and do the data support the conclusions?

Reviewer #1: Yes

2. Has the statistical analysis been performed appropriately and rigorously? 

Reviewer #1: Yes

3. Have the authors made all data underlying the findings in their manuscript fully available?

Reviewer #1: Yes

4. Is the manuscript presented in an intelligible fashion and written in standard English?

Reviewer #1: Yes

5. Review Comments to the Author

Reviewer #1: Please see the attachment.

6. PLOS authors have the option to publish the peer review history of their article (what does this mean?). If published, this will include your full peer review and any attached files.

Reviewer #1: No

---

## [Author Response · Author response to Decision Letter 0]

24 Sep 2020

This is a well written paper that I will be of use to clinicians and laboratory scientist dealing with the COVID-19 pandemic. The study is largely descriptive but is comprehensive and the information regarding days of illness, age groups and diagnostic accuracy of the Euroimmun SARS-CoV-2 ELISA is useful.

There are some problems with the English and the manuscript would benefit from an edit by a native English language speaker.

Answer: We have now edited our manuscript based on professional English editing service. We also change the title to “Antibody Responses to SARS-CoV-2 in Patients with Differing Severities of Coronavirus Disease 2019” to make it clearer.

I have some questions relating to this manuscript which I would appreciate if the authors can respond.

1. Can the authors please explain why they did not test for IgM?

Answer: Euroimmun only provides serology test for IgA and IgG, not IgM. Based on our own observation in the past that IgA seems to have pronounce responses in respiratory infection compared to IgM, we are interested in this Euroimmun system. IgA is the major isotype that important in mucosal immunity and IgA was reported to appear as early as IgM in other previously reports, we think that IgA can be used interchangeably with IgM and might even be a better marker.

2. Why did the authors choose to use their own severity score when there are already multiple severity scores available. Can the authors please comment on the limitations relating to the use of their own severity score. 

A comparison of COVID-19 severity scores is contained in the following citation. Guohui Fan, Chao Tu, Fei Zhou, Zhibo Liu, Yeming Wang, Bin Song, Xiaoying Gu, Yimin Wang, Yuan Wei, Hui Li, Xudong Wu, Jiuyang Xu, Shengjin Tu, Yi Zhang, Wenjuan Wu, Bin Cao European Respiratory Journal 2020; DOI: 10.1183/13993003.02113-2020

Answer: We are thankful to the reviewer for pointing this out. According to the severity scores that mention in Fan G et al study, CURB-65 scores based on confusion status, urea level, respiratory rate, blood pressure and age for triage patients (outpatient care, inpatient vs observation admission and inpatient admission with consideration for ICU). Pneumonia severity index (PSI) score based on age, characteristic, co-morbidity, examination and investigation lab (arterial pH, Urea, Na, Glucose, Hct, PaO2 or SaO2 and pleural effusion). SMART-COP score for pneumonia severity based on age, CXR result, albumin, vital sign, conscious status, blood gas. We did not use these criteria because the approach and test that we used are different. We admit all confirmed COVID-19 cases in the hospital as in-patient but limit the investigations in some patients when their condition is good as the transmission prevention policy in our hospital. However, when we analyzed the patient in severe group who have complete data, most of them were in severe risk group or highest risk group in CURB-65, moderate or high risk in SMART-COP and moderate to high risk from PSI score.

For a practicality standpoint, we used the severity classification based on the COVID-19 guideline for management and treatment from the Department of Medical Service, Thai Ministry of Public Health. We classified the confirmed COVID-19 patients to (1) mild symptoms; upper respiratory infection, (2) moderate; pneumonia without hypoxia and (3) severe pneumonia with hypoxia SpO2<95% , of which antiviral treatment was given. 

We added in the sentence line 115 under the “Patient population” that Classification of the confirmed case was as follows, according to the COVID-19 management guideline of the Thai Ministry of Public Health: 1) mild – asymptomatic or upper respiratory tract infection (URI), 2) moderate – pneumonia without hypoxia, and 3) severe – pneumonia with hypoxia, of which the antiviral treatment was given.

3. The authors note that they used both plasma and serum for this study. Is there any difference between plasma and serum in the ELISA? 

Answer: These serology test kit from Euroimmun has been validated from the company and stated in the leaflet that both plasma and serum can be used interchangeably. 

4. The false positives are interesting. In line 177 the authors state “From 20 serum specimens  collected from patients with other infections revealed two samples with both IgA and IgG cross-reactivity with CMV and EBV positive samples”. Can the authors prove that this is cross-reactivity or is it true positivity due to across reaction with another coronavirus? 

Answer: We did not have serology result of other coronavirus so we cannot answer this question exactly. However, based on results from other studies summarized in S2 Table, there were reports of false positive with EBV and CMV seropositive (Montesinos et al., Journal of Clinical Virology; Elslande et al., Clinical Microbiology and Infection). We have added the limitation and discussion about this point into the manuscript as well.

5. In line 175 the authors state “It should be noted that the positive results in these patients might be the result of either false positive or true positive cases with negative RT-PCR. However, we did not have any evidence to support the COVID-19 infection in these patients”. This is conjecture and should be moved to the discussion section.

Answer: We have moved this part to the discussion part as advice.

6. Given the issues relating to cross-reactivity or false positivity with SARS-CoV-2 serology – would it be possible to provide more information regarding the false positive results. The information provided is inadequate. 

Answer: The specificity was not our main objective in this study. However, we have added a paragraph regarding the specificity and cross-reactivity in the discussion part.

---

## [Editor Report · Decision Letter 1]

29 Sep 2020

Antibody Responses to SARS-CoV-2 in Patients with Differing Severities of Coronavirus Disease 2019

PONE-D-20-23838R1

Dear Dr.Hirankam, 

We’re pleased to inform you that your manuscript has been judged scientifically suitable for publication and will be formally accepted for publication once it meets all outstanding technical requirements.

Kind regards,

Walter R. Taylor

Academic Editor

PLOS ONE

---

## [Editor Report · Acceptance letter]

1 Oct 2020

PONE-D-20-23838R1 

Antibody Responses to SARS-CoV-2 in Patients with Differing Severities of Coronavirus Disease 2019 

Dear Dr. Hirankarn:

I'm pleased to inform you that your manuscript has been deemed suitable for publication in PLOS ONE. Congratulations! Your manuscript is now with our production department. 

Kind regards, 

on behalf of

Dr. Walter R. Taylor 

Academic Editor

PLOS ONE